# Multimorbidity, Frailty and Diabetes in Older People–Identifying Interrelationships and Outcomes

**DOI:** 10.3390/jpm12111911

**Published:** 2022-11-16

**Authors:** Alan J. Sinclair, Ahmed H. Abdelhafiz

**Affiliations:** 1Foundation for Diabetes Research in Older People (fDROP), King’s College, London WC2R 2LS, UK; 2Rotherham General Hospital Foundation Trust, Rotherham S60 2UD, UK; 3Department of Geriatric Medicine, Rotherham General Hospital, Rotherham S60 2UD, UK

**Keywords:** older people, diabetes, multimorbidity, frailty, outcomes

## Abstract

Multimorbidity and frailty are highly prevalent in older people with diabetes. This high prevalence is likely due to a combination of ageing and diabetes-related complications and other diabetes-associated comorbidities. Both multimorbidity and frailty are associated with a wide range of adverse outcomes in older people with diabetes, which are proportionally related to the number of morbidities and to the severity of frailty. Although, the multimorbidity pattern or cluster of morbidities that have the most adverse effect are not yet well defined, it appears that mental health disorders enhance the multimorbidity-related adverse outcomes. Therefore, comprehensive diabetes guidelines that incorporate a holistic approach that includes screening and management of mental health disorders such as depression is required. The adverse outcomes predicted by multimorbidity and frailty appear to be similar and include an increased risk of health care utilisation, disability and mortality. The differential effect of one condition on outcomes, independent of the other, still needs future exploration. In addition, prospective clinical trials are required to investigate whether interventions to reduce multimorbidity and frailty both separately and in combination would improve clinical outcomes.

## 1. Introduction

The global prevalence of diabetes is expected to increase from about 8.4% of the world population in 2017 to 10% in 2045 [1]. About half of the population with diabetes are above the age of 65 years and the expected increase in the prevalence is likely due to the increase in life expectancy [1]. Patients with diabetes are more likely to develop multimorbidity and frailty compared with those without diabetes [2]. The high prevalence of multimorbidity and frailty in older people with diabetes is likely due to diabetes-related complications and diabetes-associated conditions [3]. Another factor is the increased number of younger people diagnosed with diabetes who live long enough to develop other chronic conditions at older age [4]. Multimorbidity affects more that 80% of patients with diabetes and the number increases with increasing age and duration of diabetes [5]. For example, in one study, 97% of patients had at least one comorbidity, 88.5% had at least two and ≥4 comorbidities were observed in 25.3% of patients <65 years, 42.4% in patients 65–74 years of age and 48.5% in those ≥75 years old [6]. Frailty, another diabetes and ageing-related condition, has been recognised as a part of routine clinical assessment in older people with diabetes to identify patients at higher risk of adverse outcomes [7]. Multimorbidity and frailty in older people with diabetes will act as clinical markers that can distinguish biological from chronological age and can be predictive of downstream adverse clinical outcomes. However, there is little available literature that directly compare the independent effect of multimorbidity and frailty on these adverse outcomes. Whether the outcomes predicted by multimorbidity are different from those predicted by frailty is not yet clear. Therefore, this manuscript reviews the current literature of the role of multimorbidity and frailty as clinical markers of adverse outcomes in older people with diabetes and explores whether one condition predicts certain outcomes more precisely than the other. This may direct the clinician’s approach to setting priorities in clinical care.

## 2. Methods

### 2.1. Data Sources 

In this narrative review, the following databases: Google Scholar, PubMed and Embase were used for our literature search. We used Medical Subject Heading (MeSH) terms such as morbidity, comorbidity, multimorbidity, frailty, pre-frailty, frail, diabetes mellitus, older people, old age, elderly, outcomes, risk factors, physical function, cognitive function, glycaemic control, hypoglycaemia, mortality, quality of life, health care utilisation, hospitalisation, care home admission, adverse events and predictors as individual words and combined phrases. We reviewed the articles’ abstracts for relevance. We also manually reviewed citations in retrieved articles to identify studies that may have been overlooked in the database search. We searched Medline and Embase for articles published only in English language from 1 January 2017 to 31 December 2021. The search for articles was limited to studies that reported clear outcomes. The initial search provided 2083 articles, which were screened for inclusion criteria from titles, abstracts, full texts, or a combination of these.

### 2.2. Study Selection

Studies were included if they satisfied the inclusion criteria: 1. Studies that reported the impact of multimorbidity on the outcomes in older people with diabetes mellitus. 2. Studies that reported the impact of frailty on the outcomes in older people with diabetes mellitus. The exclusion criteria were: 1. Non-English language or non-human studies. 2. Studies with no clear outcome with clear endpoints. 3. Studies on patients without diabetes diagnosis. 4. Case reports, review articles, editorials, abstracts, conference proceedings or expert opinions. Of the 2083 studies identified, after applying exclusion criteria, a final 24 studies met the inclusion criteria and were included in this review (Figure 1).

### 2.3. Data Extraction

We independently reviewed the studies and performed data extraction in a standardised format. For each study, data were extracted in 4 main categories: 1. Author, study design, year of publication and country of origin. 2. Baseline data, which included number of patients, mean age and duration of follow up. 3. Aim of the study. 4. Main findings, which included the outcomes and end points, reported. Disagreements were resolved by consensus between authors. We have specifically looked at findings that explore the effects of multimorbidity and/or frailty on outcomes in older people with diabetes and whether certain outcomes are better predicted by one condition more than the other to help and direct future research to investigate and develop more precise predictive tools that can be more useful in every day clinical practice.

## 3. Multimorbidity

Morbidity, comorbidity and multimorbidity are commonly used terms in clinical practice. Morbidity means one illness or one disease. Comorbidity is more than one illness or disease, while multimorbidity is more than two illnesses or diseases occurring in the same person at the same time [8]. The prevalence of multimorbidity is likely to increase due to the increasing ageing of the population and the improved detection of diseases. Multimorbidity is closely linked with age and at least 50% of individuals ≥65 years of age have multimorbidity [9]. For example, the prevalence of multimorbidity is >60% in Medicare beneficiaries and >80% in adults >85 years old [10]. Common comorbid conditions in older people include cardiovascular, respiratory, cerebrovascular, musculoskeletal, endocrinal and mental health diseases. Comorbidity and multimorbidity are often used interchangeably [11]. Multimorbidity has been used to reflect the impact of multiple illnesses or diseases on the physiological function or physiologic reserve, which overlaps with the concept of frailty [12,13]. Other terms are used to reflect the impact of multimorbidity on the individual such as the morbidity burden (the overall impact of the different disease in the individual taking into account their severity) and patient’s complexity (the overall impact of the different disease in the individual taking into account their severity and other health-related attributes such as socioeconomic, cultural, environmental and patient behaviour characteristics) [11]. Attempts have been proposed to measure the burden of comorbid diseases or multimorbidity in a single multimorbidity scale. The Charlson Comorbidity Index (CCI), the Cumulative Illness Rating Scale (CIRS), the Index of Coexisting Disease (ICED), the Kaplan Index (KI) and the Incalzi index are examples [14]. Comorbidity indexes uses the current comorbid diseases then weights their pathophysiologic impact on the individual. This rating technique, which corrects for the simple additive value of the existing diseases by adding the pathophysiologic impact of these diseases, improves the overall predictive validity of the comorbidity indexes [15]. The CCI is the most widely studied multimorbidity index that positively correlates with various outcomes such as mortality, disability, readmissions and length of hospital stay [16,17,18,19]. The CIRS addresses all relevant body systems without using specific diagnoses on a five-point pathophysiologic severity scale. The CIRS has fair positive correlations for variables such as medication usage, activities of daily living (ADL), instrumental ADL (IADL) and age [20,21,22]. The ICED has a two-dimensional structure, one measuring disease severity (ICED-DS) and the other measuring the overall functional severity or disability caused by comorbidity (ICED-FS), which can be used when measuring outcomes such as disability and mortality [23]. The KI, which was specifically developed for use in diabetes research, focuses on the type of comorbidity (vascular or nonvascular) and the pathophysiologic severity (rated on a scale that ranges from zero for no or easy to control disease to three for fully decompensated disease) of the present comorbid conditions. The KI has a mortality predictive validity [24]. The Incalzi index contains 52 conditions, each weighted according to its impact on mortality risk [25]. The Incalzi age index can be computed by adding two, three or four points to the score of patients aged 76–85 years, 86–95 years and >95 years, respectively. Both the Incalzi and the Incalzi age indexes have predictive validity for mortality [25]. (Table 1).

## 4. Frailty

Frailty has been defined as a vulnerable state to psychological or physical stress factors due to diminished revere at a multiple organ level, which limits the ability to maintain homeostasis [26]. Frailty, as such, should not be seen as an inevitable part or synonymous of ageing, although its prevalence increases proportionally with age [27,28]. For example, in people older than 65 years the frailty prevalence is about 7% while in those above the age of 80 years it is around 40% [29]. Frailty is associated with a wide range of adverse outcomes such as increased risk of injurious falls, fractures, dementia, disability, reduced quality of life and pre-mature mortality [30,31,32,33,34,35]. As a result, frailty increases the burden on health care systems such as increasing attendance for emergency care, hospital admissions and residency in care homes which increases overall health costs for frail compared with non-frail older people [36,37] Therefore, frailty assessment should be integrated in routine care for older people with diabetes. A number of scales are validated for screening of frailty such as the Fried phenotype criteria, the FRAIL scale, the clinical frailty scale (CFS), the electronic frailty index (eFI) and the 35-items Rockwood frailty index [38,39,40,41,42]. The Fried criteria has been validated, in the cardiovascular health study, to predict poor mobility, risk of falls, disability in ADL, risk of hospital admission and mortality [38]. The FRAIL scale is a simple tool that does not require any testing. It includes 5 questions about Fatigue (in daily activities), Resistance (such as difficulty in climbing stairs), Ambulation difficulties, Illness history and un-intentional weight Loss [39]. Similar to Fried criteria, FRAIL scale is validated to predict the risk of disabilities in ADL and IADL and mortality [43]. The CFS is a pictographic tool that uses 9-points scale to grade functional ability and degree of frailty, based on function, and validated to predict the risk of mortality [40]. It is a practical tool that be easily used in clinical practice to predict several clinical outcomes such as length of hospital stay, falls and overall function [44]. The comprehensive geriatric assessment (CGA) is the basis for both the eFI and the 35-items Rockwood frailty index. Advantage of these indexes is that can be used for large size population sample in the primary care setting using system-integrated software [41,42]. Both scales have good predictive ability for risk of adverse clinical outcomes such as hospital admission, institutionalisation and mortality [41,42] (Table 1).

## 5. Multimorbidity and Frailty: Identifying Early Differences

Although multimorbidity and frailty are distinct conditions, they overlap and the terms are sometimes used interchangeably. Frailty can be viewed as the clinical manifestation of the progressive accumulation of physiologic decline of multiple organ systems because of increasing age and aggregation of organ dysfunctions or diseases over the years. This process is likely to be gradual starting with subclinical physiologic declines in various organ systems until organ dysfunction reaches a threshold to be clinically detectable and increases the risk of adverse outcomes. Fundamental to this process, is the multiorgan dysfunction concept, rather than a single organ, which leads to vulnerability of the individual to stressors and progression to frailty. While multimorbidity will present with signs and symptoms that reflect the underlying diseases, frailty is characterised by multiple manifestations that include weakness, wasting, weight loss, loss of endurance, reduced mobility, lack of balance, tendency to fall and slowness in performing activities. Similar to a clinical syndrome, no single manifestation is enough or essential to diagnose frailty. These characteristics are unique to frailty and it has been reported that weakness does not appear to be associated with multimorbidity [45]. Both frailty and multimorbidity lead to disability and even mortality (Figure 2). The Disablement Process Model (DPM) comprises a pathway from disease to disability, which incorporates four sequential stages starting from the disease leading to impairment, to functional decline, then to disability [46]. Different cluster of comorbid diseases may have synergistic effect and lead to a particular decline in functional abilities. For example, three multimorbidity patterns {musculoskeletal/somatic (MSO), neurological/mental health (NMH) and cardiovascular (CV)} were reported in the Australian Longitudinal Study of Women’s Health to be differentially predictive of future functional decline in ADL and IADL in older women aged 76–81 years [47]. Compared with the reference group (lowest tertile scores), women with a high score for the CV pattern morbidity had significantly worse declines in ADL, while those with NMH patterns morbidity had the greatest functional declines in IADL after 3 years of follow up [40]. Other study found that the umber, regardless of severity or heterogeneity, of the multimorbid conditions lead to disability in ADL and IADL. The odds ratio (OR) for ADL disability was 1.53 for one morbidity and 5.61 for ≥4 morbidities [48]. Additionally, geriatric conditions as opposed to other chronic conditions may have more impact on the progression to disability. The Survey of Health and Living Status of the Elderly in Taiwan (2003 and 2007) found that both multimorbid chronic health conditions and geriatric conditions (such as cognitive impairment, depressive symptoms, falls, urinary incontinence and pain) to be associated with incident disability in the young-old (65–79 years). However, only geriatric conditions were associated with incident disability in the old-old (≥80 years). The relative risk (RR) was 2.38 for 1 geriatric condition and 4.76 for ≥2 geriatric conditions, compared to no geriatric conditions [49]. Age may be a modifying factor for the effect of multimorbidity on function. For example, a 70-year-old participant with no diseases had only 0.89 limitation in physical function but this increased to 1.72 when 1 disease and 3.82 when ≥3 diseases were present. The increasing disease-induced impairments and deterioration of compensatory mechanisms with increasing age, may explain how age modifies the association between morbidity and function [50]. In summary, an increasing number of multimorbid conditions increases the risk of physical functional decline and disability, although various discrete combinations of multimorbid conditions may have different effects on the risk of disability and physical functional limitations (Figure 2).

## 6. Effects of Diabetes on Multimorbidity and Frailty

Diabetes is associated with increased loss of skeletal muscles, weakness and accelerated ageing process that leads frailty [51]. Hyperglycaemia is associated with poor muscle quality, muscle mass loss and reduced physical performance [52,53]. Several studies have demonstrated the speedy loss of muscle mass, muscle quality, muscle strength and functional capacity including a decline in gait speed in older people with compared to those without diabetes [54,55,56,57,58]. The risk of frailty appears to increase proportionally with increasing blood glucose levels. The Beijing longitudinal study of ageing II (BLSA-II) reported higher prevalence and incidence of frailty in older people, mean (SD) age 70.5 (7.8) years at baseline, with diabetes compared to those without diabetes diagnosis (19.3% vs. 11.9% and 12.3% vs. 7.0%, respectively) among a total of 10,039 participants. Among people with pre-diabetes, the prevalence of frailty (11.43%) was similar but the incidence was slightly higher (8.7%) than people without diabetes. This suggests that pre-diabetes may be a mediator to frailty [59]. Other factors, common in older people, include inadequate nutrition, especially poor protein intake, reduced physical exercise and decline in neuromuscular junction may contribute to muscular weakness and development of frailty [60]. Patients with diabetes and weak muscles are at increased risk of falls, fractures and further deterioration in physical functions, which may set a viscous circle to frailty [61]. In addition to the direct relationship of diabetes and frailty, multimorbidity associated with diabetes, especially diabetes-related complications, appear to be a mediator to frailty. The multimorbidity mediator effect to frailty also appear to be synergistic when more than one morbidity coexist with diabetes. For example, the Mexican Health and Nutrition Survey, which included 7164 older participants, mean (SD) age 70.6 (8.1) years, demonstrated that diabetes was independently associated with frailty (coefficient 0.28, *p* < 0.001). Comorbid hypertension (0.63, *p* < 0.001) or any diabetes-related complication (0.55, *p* < 0.001) incrementally increased the risk [62]. Similarly, the Japanese cross-sectional study, which included 9606 participants (≥65 years), demonstrated an increased risk of frailty {odds ratio (OR) 1.83, 95% confidence interval (CI) 1.01 to 3.45} in participants with renal impairment {estimated glomerular filtration rate (eGFR) <30.0 mL/min/1.73 m^2^} compared with participants with better renal function (eGFR ≥ 60.0 mL/min/1.73 m^2^). History of hypertension or diabetes mellitus increased the risk of frailty and the risk increased further when both conditions co-exist (OR 3.67, 95% CI 1.13 to 14.05) [63].

## 7. Effects of Multimorbidity and Frailty on Diabetes

As diabetes increases the risk of multimorbidity and frailty, the latter two are associated with increased risk of adverse outcomes in older people with diabetes. Through our literature search and after application of exclusion criteria, a total of 24 studies investigated the effect of multimorbidity and frailty on the outcomes in older people with diabetes and were included in this manuscript.

### 7.1. Effects of Multimorbidity

Twelve studies investigated the association of multimorbidity and diabetes outcomes (Table 2). Heikkala et al., reported that multimorbidity was associated with achievement of glycaemic and LDL treatment targets. However, this was a cross-sectional study which did not reflect a cause-and effect relationship and the findings were just an indication that clinicians focused on patients with multimorbidity to achieve targets more than on patients with diabetes as a single disease [64]. Umeh et al., have shown that multimorbidity is associated with poor self-rated health in a proportionate manner and this association was unconnected to glycaemic control [65]. Certain multimorbidity combinations especially those that include depression, hypertension and arthritis increased the risk of disability in older people with diabetes as demonstrated by McClellan et al., while Coles B et al., in their large retrospective analysis found that, in addition to the level of multimorbidity, cardiovascular multimorbidity increased the risk of subsequent cardiovascular events, mortality and cardiovascular mortality [66,67]. However, the effect of multimorbidity on mortality may be affected by the ethnicity of the population studied. For example, data from the UK Biobank (a population predominantly of European origin), showed that a combination of coronary heart disease and heart failure, while the Taiwan National Diabetes Care Management Program (a population predominantly of Chinese ethnicity), showed that a combination of painful conditions and alcohol problems to be associated with the largest effect size on mortality, respectively. Although the UK cohort tended to have higher body weight than that of the Taiwanese cohort, which may increase their cardiovascular risk, {median (IQR) body mass index 30.8 (27.7, 34.8) kg/m^2^ vs. 25.6 (23.5, 28.7) kg/m^2^, there is still a need for further exploration of the effects of different patterns of multimorbidity on outcomes across different ethnic groups as suggested by Chiang et al. [68]. Increased risk of emergency department visits and hypoglycaemia-related hospitalisation are another multimorbidity-related outcomes which increases in proportion with the number of morbidities as demonstrated by McCoy et al. [69]. Another retrospective report by McCoy et al., demonstrated that HbA1c levels declined as the number of comorbidities increased reflecting clinical practice of tighter glycaemic control in multimorbid patients, rather than a direct relationship between multimorbidity and glycaemic control [70]. Similarly, Chiang et al., have demonstrated no association between multimorbidity and glycaemic control in their large cross-sectional general practice study [71]. Wong et al., found that health-related quality of life was impaired with increasing number of morbidities [72]. Mental health conditions such as depression, schizophrenia, substance use disorder and anxiety, which was present in 1 in 5 of older people with diabetes, was also associated with increased risk of mortality and hospital services use as reported by Guerrero Fernández de Alba et al. in their retrospective analysis [73]. Chiang el al, found no association between multimorbidity-related adverse outcomes and glycaemia markers such as HbA1c, glycaemic variability or time blood glucose in normal range suggesting that other factors, rather than dysglycaemia, contribute to the adverse outcomes associated with multimorbidity in older people with diabetes [74]. Among comorbidities, Quiñones et al., found that the presence of depressive symptoms or stroke, in particular, pose a substantial functional burden and contributed more to disabilities in ADL and IADL in older people with diabetes than other conditions [75].

### 7.2. Effects of Frailty

Twelve studies investigated the association of frailty and diabetes outcomes (Table 3). Hanlon et al., analysed a large UK Biobank data (20,566 participants) using two frailty and two multimorbidity measures and found that each measure was associated with mortality, major adverse cardiovascular events (MACE), hypoglycaemia and fall or fracture [76]. Data from the Look AHEAD clinical trial showed that, after 8 years, the increases in frailty and multimorbidity were associated with poor cognitive function, physical function and increased mortality as reported by Espeland et al. [77]. Results From the ADVANCE trial showed that frailty predicted macro and microvascular events, all-cause mortality, cardiovascular mortality and hypoglycaemia as demonstrated by Nguyen et al., who concluded that frailty attenuated the benefits from blood pressure lowering and intensive glycaemic control [78]. In a retrospective analysis by Sable-Morita et al., frailty was a predictor of hospitalisations, institutional admissions, emergency outpatient visits, fractures, and mortality in older people with diabetes [79]. Ferri-Guerra et al., prospectively demonstrated association of frailty with all-cause hospitalisation and mortality independent of comorbidity [80]. Gual et al., investigated the impact of frailty on the outcome of a very old (≥80 years) cohort with diabetes and acute coronary syndrome. The association between diabetes and outcomes (incidence of death or readmission after 6 months) was not significant in robust patients, but it was significant in frail patients [81]. Among patients with diabetic kidney disease, frailty increased the risk of progression to end stage renal disease on a dose–response relationship and mortality, compared to those without frailty as reported by Chao et al. [82]. The prospective study by Kitamura et al., showed that all-cause mortality and disability in older people with mild diabetes were strongly affected by the presence of frailty [83]. Frailty was associated with low health related quality of life, depression, lean body mass and higher numbers of health-care visits in people with diabetes and chronic kidney disease as demonstrated in a cross-sectional analysis by Adame Perez et al. [84]. Chao et al., showed that both pre-frailty and frailty were associated with increased mortality, cardiovascular events, hospitalisation and health care utilisation [85]. In the community prospective study by Thein et al., frailty was associated with disability, which was potentiated by the presence of cognitive impairment. In addition, frailty, cognitive impairment or both were strong predictors of mortality [86]. In another community study by Li et al., frailty was associated with increased hospitalisation while both pre-frailty and frailty were associated with increased emergency department visits [87].

## 8. Discussion

It appears, from the above studies, that the effects of both multimorbidity and frailty on outcomes in older people with diabetes are similar. (Table 4) In addition, some of these studies are cross-sectional design that showed associations rather than causation between multimorbidity or frailty and outcomes. This was mostly demonstrated in studies that examined the relationship between multimorbidity and low HbA1c. The findings of the association of multimorbidity and low HbA1c is likely reflecting a clinical practice attitude that is concerning with achieving glycaemic targets in these patients compared with patients with only diabetes and no morbidities [64,70,71].

### 8.1. Patterns of Multimorbidity

The multimorbidity studies have consistently showed that increasing level of multimorbidity is proportionally associated with adverse outcomes [65,67,69,70,72,74]. Multimorbidity can be either concordant (diabetes-related) or discordant (diabetes-unrelated). Several studies have examined the differential effect of concordant as opposed to discordant multimorbidity on outcomes. Some studies shown similar effects of concordant and discordant multimorbidity on glycaemic control [64,70,71]. Other studies have shown that both types of multimorbidity were significantly associated with mortality, however the hazard ratios were largest for the concordant multimorbidity especially cardiovascular and renal diseases [68]. Certain multimorbidity combinations may have especially significant impact on outcomes. For example, multimorbidity combinations that include depression, hypertension and arthritis were associated with increased risk of disability [66]. Cardiovascular morbidity increased the risk of cardiovascular events and mortality [67]. Mental health multimorbidity was associated with adverse outcome especially in patients with substance use disorder or schizophrenia [73]. The addition of depression or stroke to existing multimorbidity was associated with a substantial increase in functional disabilities [75]. It is however, not very clear how chronic conditions form distinct groups and patterns of co-existence that clusters in older people with type 2 diabetes. The most common multimorbidity clusters with diabetes are: 1. Cardiometabolic diseases such as obesity, hypertension and dyslipidaemia, 2. Vascular conditions such as macrovascular disease, microvascular disease, atrial fibrillation and CKD, 3. Mental health conditions such as depression, anxiety and substance abuse [88]. However, with increasing age and longer duration of diabetes, the disease burden increases as well as the prevalence of all morbidities with diversification of conditions as diabetes progresses [89]. In other words, the number and heterogeneity of clusters in the advanced diabetes depart from the original distinct patterns found in the early period of diabetes [90]. Therefore, the impact of multimorbidity clusters on outcome may be more prominent in younger age groups with short duration of diabetes while in older people with long duration of diabetes the high multimorbidity burden, rather than specific clusters, will have more impact on outcomes especially in patients with advanced disease and multiple end-organ damage [91]. However, the current evidence does not provide information about the trajectory, temporal sequence or trends of multimorbidity and clustering in patients with diabetes. In addition, the definition of multimorbidity that relies on the simple sum of individual diseases does not consider the severity of individual conditions or the interaction between morbidities and still needs further exploration [92].

### 8.2. Multimorbidity-Frailty Overlap

The differential contribution of multimorbidity and frailty to diabetes-related outcomes is not yet clear. For example, none of the multimorbidity studied, discussed above, has assessed or adjusted for the frailty. Therefore, frailty may be an unmeasured confounding factor for the outcomes associated with multimorbidity. On the other hand, some of the frailty studies did not adjust for the confounding effect of multimorbidity on frailty-related adverse outcomes [78,79,81,82,85]. Other studies reported frailty-related outcomes independent of multimorbidity. The study by Ferri-Guerra et al., has shown that frailty, independent of multimorbidity (assessed by CCI) was a predictor of all-cause hospitalisation and mortality [80]. Kitamura et al., showed that frailty predicted mortality and disability independent of a number of comorbidities including hypertension, high cholesterol, low cholesterol, CKD, overweight, underweight, anaemia, low Mini-mental examination score, history of stroke and smoking status [83]. The adverse outcomes predicted by frailty, as reported by Chao et al., was adjusted for comorbidities such as mental health illness, obesity, severity of diabetes as measured by adjusted diabetic complication severity index, smoking and alcohol abuse [85]. Thein et al.’s, report of the association of frailty with mortality was independent of some comorbidities that included cardiac disease, stroke, depression, obesity, arthritis and hip fracture [86]. The association of frailty with increased risk of emergency department visits and hospitalisation as reported by Li et al., was adjusted for ADL disability, IADL disability and history of falls [87]. Among the multimorbidity and frailty studies described in this review, only two studies examined simultaneously, the predictive effects of both conditions on outcomes. The study by Hanlon et al., have examined the effect of both multimorbidity and frailty on outcome [76]. However, this study did not report a direct effect on the outcome of one condition independent of the other. In their, post hoc analysis, they found that frailty was associated with an increased risk of mortality at each level of multimorbidity and subjects with combined frailty and multimorbidity had a greater risk of mortality than those with frailty or multimorbidity alone. This may suggest that both multimorbidity and frailty have an additive effect on diabetes-related outcomes. The Look AHEAD study, although was not designed to examine the differential effects of multimorbidity and frailty, reported an independent effect of each condition, when adjusted for each other, on functional and mortality outcomes [77]. This suggests that multimorbidity and frailty are overlapping, but not, interchangeable risk factors for adverse outcomes in older people with diabetes.

### 8.3. Identifying Interventions to Target Multimorbidity and Frailty

Clinical interventions targeted at improving multimorbidity and frailty are likely to be associated with less burden on health care resources and better health-related quality of life for older people with diabetes. However, 8 years data from the Look AHEAD study, suggests that the improvement in gait speed was independent of improvement in multimorbidity and frailty and the intensive lifestyle intervention benefits on multimorbidity and frailty did not translate into improvements in the risks of cognitive impairment or mortality [77]. Authors suggested that the magnitude of benefits in multimorbidity and frailty was too small to have a measurable effect on outcomes. Intensive lifestyle intervention, such as weight loss, may have a differential effect across age groups. For example, with advanced age and a decline in health, weight loss intervention may have a reverse benefit. Therefore, the impact of interventions targeting multimorbidity and frailty to reduce the risks of adverse outcomes are not yet clear. Higher prevalence of multimorbidity is observed in populations living in more deprived areas, which highlights the need to address health inequalities [90]. Although current health care services focus on the prevention of cardiovascular and other physical health conditions, the growing burden of mental health disorders will need service and workforce review and restructure [90]. The relationship between multimorbidity and glycaemic control appears to be mixed [93]. In addition, the increased mortality predicted by multimorbidity is not linked to HbA1c [71]. It is likely that older people with multimorbidity were treated more with insulin to achieve lower HbA1c targets with a potential high risk of hypoglycaemia and uncertain long-term benefit [70]. Similarly, the relationship between frailty and glycaemia appears to be mixed but it could be related to the differences in frailty metabolic phenotypes [94]. Therefore, better understanding of the implications of multimorbidity and frailty on glycaemic control is still required. With increasing prevalence of multimorbidity and frailty in older people with diabetes and their association with adverse outcomes, the clinical guidelines should move from a disease-specific to a patient-centred holistic care. Furthermore, this holistic approach should include the care of discordant diseases, particularly mental health conditions, in addition to the care of traditional concordant cardiometabolic diseases [95]. Guidelines also provide little direction for self-care in the presence of other chronic conditions [96]. It has been shown that diabetes-concordant comorbidities are more associated with higher adherence to diabetes self-care compared to discordant conditions therefore, guidelines should integrate diabetes self-care among patients with multimorbidity, especially discordant conditions, in order to optimise clinical outcomes [97].

## 9. Conclusions

Multimorbidity and frailty are prevalent in older people with diabetes and are associated with a wide range of adverse outcomes including disability and mortality. The number of morbidities and the severity of frailty proportionally increase the risk of adverse outcomes. The relationship between multimorbidity and frailty with glycaemic control is not clear and needs further exploration. The pattern and clustering of morbidities may have some effect on adverse outcomes prediction although this may be attenuated with increasing age and duration of diabetes, which is associated with multiple organ damage. However, it appears that the development of discordant conditions such as mental health disorders will further increase the risk of adverse outcomes including less adherence to self-care. Therefore, comprehensive diabetes care guidelines that incorporate a holistic approach that includes screening and management of discordant conditions, especially mental health disorders such as depression, is required.

## 10. Future Perspectives

Multimorbidity and frailty are associated with a wide range of adverse outcomes in older people with diabetes. However, the differential effect of one condition, independent of the other, still not clear which will need future exploration. For example, none of the morbidity studies discussed in this review have adjusted for the effect of frailty. Therefore, frailty can be unmeasured confounding factor in these studies. On the other hand, some of the frailty studies adjusted for comorbidities, which may suggest that frailty, independent of comorbidity, is a risk factor for adverse outcomes in older people with diabetes but this evidence is not substantiated or replicated across other studies. In addition, there is no studies designed to directly compare the differential effects of multimorbidity versus frailty as a predictor of adverse outcomes. Furthermore, the studies discussed in this review are mostly cross sectional or retrospective that may demonstrate some associations but not necessarily detect causations. Therefore, current evidence, at best, suggests an overlap between frailty and multimorbidity and, therefore, risk stratification in older people with diabetes using multimorbidity and/or frailty needs further research. Not all comorbid conditions have the same impact on the total multimorbidity burden. We still need to know more about the pattern and clusters of multimorbidity that is associated with the greatest risk of adverse events as well as whether this effect differs across different ethnic groups. It is still not clear whether the effect of certain combination of chronic conditions is additive or synergistic in predicting outcomes. These comorbid conditions can subsequently be used to develop a precise diabetes-specific comorbidity measures. Similarly, a specific multimorbidity pattern combined with frailty measures can be explored in order to predict adverse outcomes accurately. Research is still required to clearly distinguish between multimorbidity and frailty. For example, chronic diseases with increased catabolic state that constitute multimorbidity may have similar symptoms to frailty such as fatigue, weight loss and exhaustion, which has been termed secondary frailty, as opposed to purely age-related primary frailty [98,99]. Therefore, longitudinal studies that include frail participants with multimorbidity are needed to investigate the independent effect of each condition on adverse outcomes. This may help develop a more refined tool that incorporate multimorbidity and frailty to better predict outcomes. In addition, accumulating evidence suggests that oxidative stress plays a significant role in the pathogenesis of insulin resistance, diabetes and cardiovascular disease morbidity and eventually frailty. Research is, therefore, required to further understand the pathological processes beyond this association and to develop new preventative therapies [100]. Lastly, more information is required to investigate whether global intervention for frailty, disease specific intervention, or a combination of both is best strategy to improve outcomes.

## 11. Key Points

Multimorbidity and frailty are predictors of adverse outcomes in older people with diabetes.Whilst the pathogenesis and nature of multimorbidity and frailty may be diverse, the adverse outcomes predicted by multimorbidity and frailty are similar.Mental health disorders significantly augment adverse outcomes predicted by multimorbidity.The predictor effect of multimorbidity independent of frailty, and vice versa, still needs further clarification.Prospective clinical trials are required to investigate whether interventions to reduce multimorbidity and frailty would improve outcomes.

## Figures and Tables

**Figure 1 jpm-12-01911-f001:**
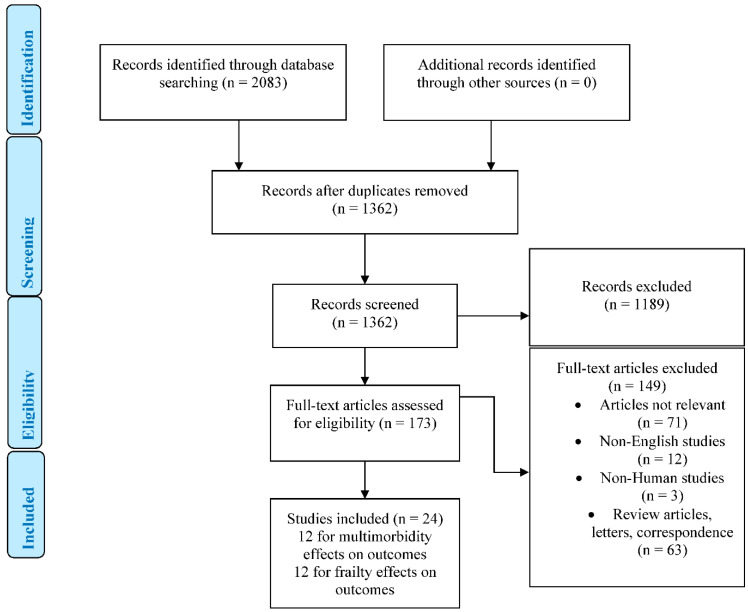
PRISMA flow diagram.

**Figure 2 jpm-12-01911-f002:**
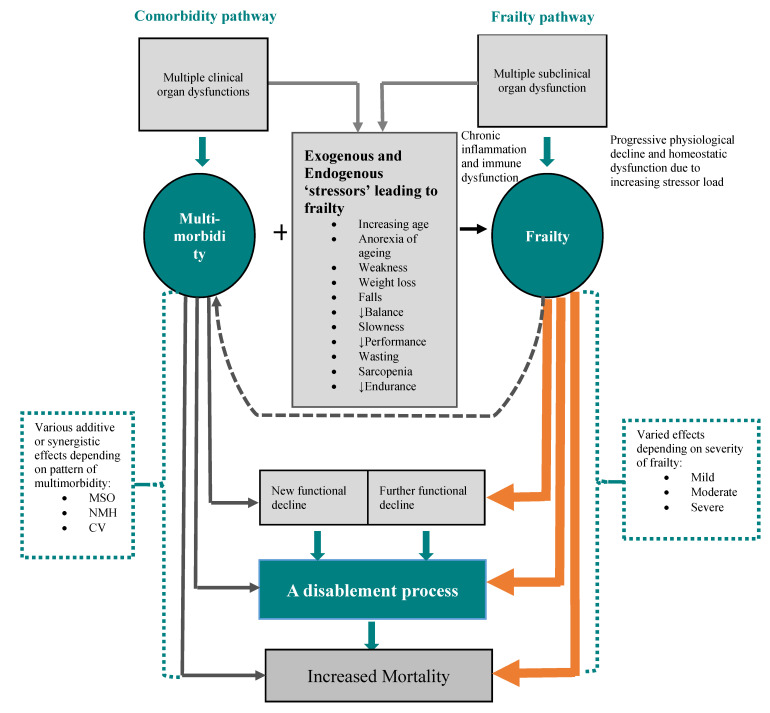
Multimorbidity-frailty interaction. Multiple subclinical organ dysfunction lead to frailty, multimorbidity associated with other factors such as weakness lead to frailty, both multimorbidity and frailty lead to disability, which in turn worsens multimorbidity and frailty and eventually increases mortality. MSO = Musculoskeletal/somatic, NMH = Neurological/mental health, CV = cardiovascular.

**Table 1 jpm-12-01911-t001:** Commonly used multimorbidity and frailty assessment indexes and tools.

**Multimorbidity**				
**Index**	**Items**	**Weights**	**Score**	**Population and Advantages**
**CCI**	19 conditions.	Range: RR 1.2–1.5 for 0 conditions to RR > 6.0 for 6 conditions.	Sum of weights.	Mixed populations including elderly, care home residents and cancer patients. Correlates with mortality, disability, readmissions and length of hospital stay.
**CIRS**	13 body systems	Ranges from 0 for no impairment to 4 for life threatening impairment.	Sum of weights.	Mixed populations including elderly, care home residents and cancer patients. Correlates with ADL, IADL and age.
**ICED**	ICED-DS 14 disease categories ICED-FS 10 functional categories.	ICED-DS 1–5. ICED-FS 1–3.	1–4	Care home residents and those with hip replacement. Predicts mortality and disability.
**KI**	Vascular or non-vascular diseases.	Ranges from 0 for no or easy to control to 3 for full decompensated disease.	According to the most severe condition.	Diabetes mellitus and breast cancer. Has a mortality predictive validity.
**Incalzi**	52 conditions.	Based on RR of mortality.	Sum of weights, adding points for every decade above age of 75 years.	Mixed populations including elderly. Has predictive validity for mortality.
**Frailty**				
**Tool**		**Criteria**		**Advantages**
**Fried criteria**		5-point scale: weight loss, exhaustion, weakness assessed by grip strength, reduced physical activity and slowness measured by gait speed.		Identifies robust (score 0), pre-frail (score 1–2) and frail (score >3) individuals but requires two practical measurements.
**FRAIL scale**		5-point scale: fatigue, resistance, ambulation, illness and loss of weight.		Can be self-assessed and does not require measurements by healthcare professionals.
**CFS**		9-point scale that describes patient’s functional characteristics and categorise them from very fit to severely frail.		Uses clinical descriptors and pictographs to stratify older people according to level of function to predict mortality or institutionalisation.
**eFI**		Uses the cumulative deficit model to identify and score frailty based on routine interactions of patients with their general practitioner.		Can be used to screen for the whole practice population who are >65 years old.
**35-Items Rockwood frailty index**		35 items, based on data from chronic diseases, disabilities in activities of daily living, cognition, nutrition, visual and hearing impairment.		Includes comprehensive data as a part of comprehensive geriatric assessment.

CCI = Charlson comorbidity index, RR = Relative risk, CIRS = Cumulative Illness Rating Scale, ADL = Activities of daily living, IADL = Instrumental ADL, ICED = Index of Coexisting Disease, ICED-DS = ICED-disease severity, ICED-FS = ICED-function severity, KI = Kaplan Index, CSF = Clinical frailty scale, eFI = electronic frailty index.

**Table 2 jpm-12-01911-t002:** Recent studies exploring effects of multimorbidity on outcomes in older people with diabetes.

Study	Patients	Aim to	Main Findings
Heikkala E et al., cross-sectional, Finland, 2021 [64].	4545 subjects with type 2 DM, mean (SD) age 70.9 (12.3) Y.	Investigate associations of multimorbidity and treatment goals, HbA1c, LDL cholesterol and SBP.	A. 93% of subjects had general, 21% concordant, 8 % discordant and 64% both multimorbidities, respectively. B. General multimorbidity, concordant multimorbidity and discordant multimorbidity significantly associated with achievement of HbA1c target (OR 1.32, 95% CI 1.01 to 1.70, 1.47, 1.10 to 1.95 and 1.32, 1.01 to 1.72, respectively). C. Similar findings with attainment of LDL target (1.34, 1.03 to 1.74, 1.33, 1.00 to 1.78 and 1.36, 1.05 to 1.78, respectively).
Umeh K, cross sectional, UK, 2021 [65].	280 subjects with type 2 DM, median age 65–74 Y.	Examine self-rated health related to multimorbidity, glycaemia and BMI.	Odds of ‘fair/bad/very bad’ increased 10-fold in patients with 3 conditions (OR 10.11, 95% CI 3.36 to 30.40) and 4 conditions (10.58, 2.9 to 38.25) irrespective of glycaemic control (*p* < 0.001).
McClellan SP et al., prospective cohort, Mexico, 2021 [66].	Total 2558 subjects with DM, 1997 with and 561 without morbidities.	Investigate relationship of combinations of morbidities and disability.	A. Top 3 combinations were diabetes-hypertension (31.9%), diabetes-hypertension-depression (19.4%) and diabetes-depression (10.6%). B. DM-hypertension-depression (IRR 2.44, CI 1.65 to 3.60), DM-depression (2.37, 1.34 to 4.21) and DM-hypertension-arthritis-depression (3.74, 2.08 to 6.73) associated with higher ADL-IADL scores.
Coles B et al., retrospective, UK, 2021. [67]	Total 120,409 subjects with type 2 DM, mean (SD) age 63.5 (13.4) Y.	Quantify risk of CVD events, all-cause mortality and CV mortality in DM and multimorbidity.	A. Compared with DM only, ≥4 morbidities increased risk of CV events (HR 2.57, 95% CI 2.45 to 2.69), all-cause mortality (1.73, 1.68 to 1.78) and CV mortality (2.68, 2.52 to 2.85). B. Compared with no CVD morbidity, ≥2 morbidities increased risk of CV events (2.42, 2.35 to 2.49), all-cause mortality (1.44, 1.42 to 1.47) and CV mortality (2.44, 2.35 to 2.54).
Chiang JI et al., longitudinal cohort, UK-China, 2020 [68].	UK Biobank, 20,569 subjects, mean (SD) age 60.2 (6.8) Y, Taiwan NDCMP 59,657 subjects, mean (SD) age 60.8 (11.3) Y.	Explore associations of multimorbidity with baseline HbA1c and all-cause mortality in type 2 DM.	Increasing total and discordant multimorbidity were associated with lower HbA1c and increased mortality in both datasets. A. In UK Biobank, HRs (95% CI) for all-cause mortality in people with 1, 2, 3 and 4 morbidities compared with no morbidities were 1.20 (0.91 to 1.56), 1.75 (1.35 to 2.27), 2.17 (1.67 to 2.81) and 3.14 (2.43 to 4.03), all *p <* 0.001. B. HRs for mortality in Taiwan NDCMP were similar. C. Largest effect size on mortality was CHD and HF in UK Biobank (HR 4.37, 95% CI 3.59 to 5.32) *p <* 0.001, and painful conditions and alcohol in Taiwan NDCMP (4.02, 3.08 to 5.23) *p <* 0.001.
McCoy RG et al., cohort, US, 2020 [69].	201,705 subjects with DM, mean (SD) age, 65.8 (12.1) Y.	Examine associations of multimorbidity and other factors with hypoglycaemia-related ED visits and hospitalisations.	Risk of hypoglycaemia-related ED visits and hospitalisations increased by number of comorbidities (IRR of 1.66, 95% CI 1.42 to 1.95) in the presence of 2 comorbidities to IRR of 4.12, 3.07 to 5.51 with ≥8 comorbidities compared with ≤1 morbidity.
McCoy RG et al., retrospective, US, 2020 [70].	194,157 patients with type 2 DM, mean (SD) age 66.2 (11.7) Y.	Examine impact of DM-concordant, discordant and advanced morbidities on HbA1c.	A. 45.2% patients had DM-concordant, 2.7% discordant, 30.6% both morbidities and 13.0% had ≥1 advanced morbidities. B. Mean (SD) HbA1c was highest in patients with no comorbidities, 7.4% (1.7), slightly lower in those with concordant, 7.3% (1.5), much lower in those with discordant, 7.1% (1.5), both, 7.1% (1.4) and advanced comorbidities, 7.0 (1.3). C. In patients with discordant comorbidities, HbA1c declined as number of comorbidities increased, 7.1% (1.6) with 1 to 6.6% (1.2) with ≥3 morbidities.
Chiang JI et al., cross sectional, Australia, 2020 [71].	69,718 subjects with type 2 DM, mean (SD) age 66.42 (12.70) Y.	Explore prevalence of multimorbidity and its association with HbA1c.	A. >90% of participants had multimorbidity, 83.4% discordant and 69.9% concordant conditions. B. Top 3 discordant were painful diseases (55.4%), dyspepsia (31.6%), depression (22.8%) and concordant were hypertension (61.4%), CHD (17.1%) and CKD (8.5%). C. No association of multimorbidity and HbA1c.
Wong FLY et al., cross sectional, China, 2020 [72].	2326 patients with DM, 60% aged ≥65 Y.	Estimate health scores by sociodemographics.	Patients with ≥3 morbidities are more likely to show a lower health-related quality of life scores than those with DM alone.
Guerrero Fernández de Alba I et al., retrospective, Spain, 2020 [73]	63,365 subjects with type 2 DM, mean (SD) age 69.9 (12.1) Y.	Study mental health comorbidity prevalence and its association with outcomes.	Mental health multimorbidity prevalent in 19% of subjects and increased mortality risk (OR 1.24, 95% CI 1.16 to 1.31), all-cause hospitalisation (1.16, 1.10 to 1.23), DM-related hospitalisation (1.51, 1.18 to 1.93) and emergency room visits (1.26, 1.21 to 1.32).
Chiang JI et al., cross-sectional, Australia, 2020 [74].	279 subjects with type 2 DM, mean (SD) age 60.4 (9.9) Y.	Explore associations of multimorbidity and HbA1c, GV and TIR.	A. 89.2% of subjects had multimorbidity. B. Most prevalent was hypertension (57.4%), painful conditions (29.8%), CHD (22.6%) and depression (19.0%). C. Multimorbidity was not associated with HbA1c, GV or TIR.
Quiñones, AR et al., prospective cohort, US, 2019 [75].	3841 subjects with DM, mean (SD) age 68.1 (9.5) Y.	Identify multimorbidity combinations and their association with poor functional status.	Depressive symptoms or stroke, added to DM-multimorbidity combinations associated with higher ADL-IADL limitations: A. DM-arthritis-hypertension-depressive symptoms vs. DM-arthritis-hypertension: IRR 1.95, 95% CI 1.13 to 3.38). B. DM-arthritis-hypertension-stroke vs. DM-arthritis-hypertension: (2.09, 1.15 to 3.82).

DM = Diabetes mellitus, SD = Standard deviation, Y = Years, SBP = Systolic blood pressure, OR = Odds ratio, CI = Confidence interval, BMI = Body mass index, ADL = Activities of daily living, IADL = Instrumental ADL, CVD = Cardiovascular disease, HR = Hazard ratio, NDCMP= National Diabetes Care Management Program, CHD = Coronary heart disease, HF = Heart failure, ED = Emergency department, IRR = Incidence rate ratio, CKD = Chronic kidney disease, GV = Glycaemic variability, TIR = Time in range.

**Table 3 jpm-12-01911-t003:** Recent studies exploring effects of frailty on outcomes in older people with diabetes.

Study	Patients	Aim to	Main Findings
Hanlon P et al., longitudinal cohort, UK, 2021 [76].	UK Biobank, 20,566 with type 2 DM aged 40–72 Y.	Assess implications of frailty/multimorbidity in middle/older-aged people with type 2 DM using 2 morbidity and 2 frailty measures.	A. 42% of participants were frail or multimorbid by at least one measure, 2.2% by all four measures. B. Each measure was associated with mortality, MACE, hypoglycaemia, fall or fracture. C. Mortality risk was higher in older vs. younger participants with a given level of frailty (1.9%, and 9.9% in men aged 45 and 65, respectively or multimorbidity (1.3% and 7.8% in men with 4 morbidities aged 45 and 65, respectively).
Espeland MA et al., prospective, US, 2021 [77].	3842 subjects with type 2 DM aged 45–76 Y at baseline, F/U 8 Y.	Examine effect of multimorbidity and frailty on cognition, physical function and mortality.	Increases in both multimorbidity and frailty index were associated with poor composite cognitive function and 400 m walk speed and increased risk for death (all *p* < 0.001).
Nguyen Tu N et al., retrospective, multicentre, 2021 [78].	11,140 subjects with type 2 DM, mean (SD) age, 65.78 (6.39) Y.	Explore effect of frailty on intensive glycaemic and blood pressure control.	A. Frailty increased risk of combined macro- and microvascular events (HR 1.03, 95% CI 0.90 to 1.19, *p* = 0.02) and all-cause mortality (1.11, 0.92 to 1.34). B. Severe hypoglycaemia was higher in frail, 8.39 (6.15 to 10.63) vs. 4.80 (3.84 to 5.76) in non-frail (*p* < 0.001). C. No significant difference in discontinuation of BP treatment due to hypotension/dizziness between frail and non-frail.
Sable-Morita S et al., retrospective, Japan, 2021 [79].	477 subjects with DM, mean (SD) age 74.2 (6.2) Y.	Assess whether frailty and DM-related factors could predict occurrence of adverse events.	Microvascular complications and frailty were significant predictors of adverse event incidence, respective OR (95% CI) 1.403 (1.11 to 1.78) per additional complication, 2.419 (1.33 to 4.40) for frailty; both *p* < 0.05).
Ferri-Guerra J et al., retrospective, US, 2020 [80].	763 subjects with DM, mean (SD) age 72.9 (6.8) Y.	Determine association of frailty with all-cause hospitalisations and mortality.	Frailty was associated with higher all-cause hospitalisations, HR 1.71 (95% CI 1.31 to 2.24), *p* < 0.0001 and greater mortality, 2.05, 1.16 to 3.64), *p* = 0.014.
Gual M et al., prospective, Spain, 2019 [81].	Total 532 subjects with ACS, 212 with DM, mean (SD) age 83.7 (5.0) Y.	Evaluate impact of DM on mortality or 6-month readmission according to frailty status.	Association of DM and incidence of clinical outcomes was significant only in patients with established frailty (HR 1.72, 1.05 to 2.81) compared to non-frail patients.
Chao CT et al., retrospective, Taiwan, 2019 [82].	165,461subjects with DKD, aged >20 Y.	Examine effect of frailty on DKD progression to ESRD, mortality, and adverse episodes.	A. Subjects with 1, 2, and ≥3 on FRAIL scale had increased risks of ESRD and mortality HRs 1.13, 1.18, and 1.2 and 1.25, 1.41, and 1.34, respectively. B. frailty increased risk of CV events and ICU admission in a dose response-manner.
Kitamura A et al., prospective, Japan, 2019 [83].	1271 subjects, 174 with DM, mean (SD) age 71.0 (5.6) Y, F/U 8.1 Y.	Clarify risks of death and disability in diabetes, frailty, both or neither.	A. Compared with non-frail subjects without diabetes, those with diabetes and frailty had higher risks of mortality, HR 5.0, 95% CI 2.4 to 10.3) and incident disability (3.9, 2.1 to 7.3). B. Non-frail with diabetes did not have a significant increased risk of mortality, but a tendency for disability compared with non-frail without diabetes.
Adame Perez SI et al., cross-sectional, Canada, 2019 [84].	41 subjects with DM and CKD, median (range) age 70 (65–74) Y.	Compare differences in body composition, HRQoL, mental health, cognition and vitD status with health-care utilization by frail and non-frail.	Frail, compared with non-frail, subjects had lower lean body mass, lower HRQoL scores, more depression (*p* = <0.05) and higher numbers of health visits (*p* < 0.05). No differences in health-care visit types or vitD status were noted between frail and non-frail participants.
Chao CT et al., longitudinal cohort, Taiwan, 2018 [85].	560,795 subjects with type 2 DM, mean (SD) age 56.4 (13.8) Y, 3.14 Y F/U.	Examine frailty impact on long-term mortality, CV risk, all-cause hospitalisation, and ICU admission.	Pre-frailty (1, 2 FRAIL scale) and frailty (≥3) increased risk of: A. Mortality, HR 1.05, 1.13, and 1.25 (95% CI 1.02 to 1.07, 1.08 to 1.17 and 1.15 to 1.36, respectively). B. CV events, 1.05, 1.15, and 1.13 (1.02 to 1.07, 1.1 to 1.2 and 1.01 to 1.25, respectively). C. Hospitalisation, 1.06, 1.16, and 1.25 (1.05 to 1.07, 1.14 to 1.19, and 1.18 to 1.33, respectively). D. ICU admission, 1.05, 1.13, and 1.17 (1.03 to 1.07, 1.08 to 1.14, and 1.06 to 1.28, respectively) compared to non-frail.
Thein FS et al., prospective, Singapore, 2018 [86].	2696 subjects, 486 with DM, mean (SD) age 67.3 (7.5) Y.	Investigate effect of frailty and cognitive impairment on functional and mortality outcomes.	A. Frailty associated with higher prevalence of IADL disability, OR 6.72, 95% CI 1.84 to 24.5. B. Frailty and cognitive impairment associated with highest prevalence of IADL (17.8, 3.66 to 8.68) and ADL disabilities (93.8, 23.6 to 372.4). C. Cognitive impairment (HR 2.72, 95% CI 1.48 to 5.01), frailty (4.30, 1.88 to 9.82) and cognitive impairment with frailty (8.41, 3.95 to 17.9) associated with mortality.
Li CL et al., cross-sectional, Taiwan, 2018 [87].	3203 subjects, 719 with DM, aged ≥ 65 Y.	Investigate prevalence of frailty and its relationship with health care.	A. Frailty, but not pre-frailty, significantly associated with hospitalisation, OR 5.31, 95% CI 1.87 to 15.10). B. Pre-frail and frail significantly associated with emergency department visits (2.64, 1.35 to 5.17 and 4.05, 1.31 to 12.49, respectively).

DM = Diabetes mellitus, Y = Year, MACE = Major adverse cardiovascular events, F/U = Follow up, SD = Standard deviation, HR = Hazard ratio, CI = Confidence interval, BP = Blood pressure, OR = Odds ratio, DKD = Diabetic kidney disease, ESRD = End stage renal disease, CV = Cardiovascular, ICU = Intensive care unit, HRQoL = Health-related quality of life, vitD = vitamin D, IADL = Instrumental activities of daily living.

**Table 4 jpm-12-01911-t004:** Similar outcomes predicted by multimorbidity and frailty in older people with diabetes.

Multimorbidity	Frailty
Low HbA1c Poor self-rated health Disability Cardiovascular events Cardiac mortality All-cause mortality Poor health related quality of life Emergency department visits Hypoglycaemia-related hospitalisation All-cause hospitalisation Falls Fractures Poor cognitive function Poor physical function	Hypoglycaemia Mortality Falls Fractures Cardiovascular events Poor cognitive function Poor physical function Institutional admissions Microvascular complications Hospital readmission Progression to ESRD in DKD Poor health related quality of life Depression Frequent health care visits Lean body mass

ESRD = End stage renal disease, DKD = Diabetic kidney disease.

## Data Availability

Not applicable.

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
