# Peer review of "Multimorbidity, Frailty and Diabetes in Older People–Identifying Interrelationships and Outcomes"

_jpm, 2022, doi:10.3390/jpm12111911_

Round 1
Reviewer 1 Report
Reviewing literatures of interrelationships between mutimorbidity, frailty and diabetes and their adverse outcomes in older people is an interesting topic. But this paper needs massive revision to be published.
1. From [3. Multimorbidity] to [6. Effects of diabetes on multimorbidity and frailty] is more rather conceptual frame work rather than literature review.
If would be more appropriate to be replaced after [1. introduction]
2. The results of this study is [7. Effects of multimorbidity and frailty on diabetes] . Please organize in orders and describe more clearly so that the readers can understand.
3. The author described that the literature seach was done following PRISMA. But the MsSH terms that the authors suggested are not using PI strategy. Therefore, the reveiwer can not be sure of sensitivity(validity) of 24 literature outcomes. Please submit search strategy table as an Appendix.
Author Response
Many thanks for your comments and suggestions for the manuscript.
- We agree with you, the first part of the manuscript is an introductory section rather than the main topic. We have divided it into subtitles, rather than including the whole thing as an introduction, for ease of reading. However, we are happy with any more suitable format suggested by yourself and the editorial team.
- The result of this study is....etc section 7. We are not very clear about what is required in this point but we have looked at the order of the studies in text and in tables and made sure order is matching and each study finding is explained as clear as we could understood it.
- We agree with your comments in this point but we have made it explicit that this manuscript is not a systematic review (highlighted in the methods section). It is a narrative review and we just used the PRISMA flow chart for ease of reading but we did not follow a systematic review strategy. The heterogeneity of the studies with very different outcomes and the lack of randomised controlled trials made it not possible to systematically review the literature and draw a statistically valid causal relationship between morbidity or frailty and specific adverse outcomes. This is a limitation of the current evidence and this has been highlighted in the manuscript and we concluded that future research is this area is required. We are happy to delete Figure 1 if you think it can cause confusion.
Reviewer 2 Report
My compliments with the authors, diabetes is important in older patients for the comorbility that they could present through the years. I believe that adding also the oxidative stress as a cause that can lead to the comorbilities is important and they can add the article with tile
The molecular link between oxidative stress, insulin resistance, and type 2 diabetes: A target for new therapies against cardiovascular diseases by andreadi et al. Minor check of the english
Author Response
Many thanks for your comments and suggestions. We have added oxidative stress and cited the suggested reference to the future perspective section (highlighted in the manuscript) as an important factor that needs further exploration in future research.
Reviewer 3 Report
This is a very good retrospective study in which the author has evaluated the role of multimorbidity and fragility in patients with diabetes. The figures and tables are helpful.
The relationship between diabetes and mental health illness is well established, and it is well known that patients with diabetes and depression have poorer outcomes than those without depression.
I would recommend the author be more objective if possible and consider using a scoring system to predict the likelihood of mental health illness in patients with diabetes with other comorbid conditions. This can help clinicians to be more proactive in moderate to high-risk cases.
Author Response
Many thanks for your comments and suggestions for the manuscript. We fully agree with you that mental health morbidities, especially depression, is associated with poor outcomes. This has been highlighted throughout the manuscript.
Developing a scoring system to predict the impact of mental health on diabetes outcomes sounds as a very interesting idea, which we would like to develop in a separate and more specific article on that issue in the near future.
We think the idea is however, slightly outside the scope of this manuscript which focuses mainly on multimorbidity and frailty as a general issue and not specifically on mental health morbidity.
Round 2
Reviewer 1 Report
Thank you very much for well organized revised paper.
I would like to recommend small comments.
The author have extracted 24 articles by literature search and these articles are discribed in [7.Effects of multimorbidity and frailty on diabetes] because search artucle inclusion criteria were: 1. Studies that reported the impact of multimorbidity on the outcomes in older people with diabetes mellitus. 2. Studies that reported the impact of frailty on the outcomes in older people with diabetes mellitus.
From [3. Multimorbidity] to [6. Effects of diabetes on multimorbidity and frailty] is the narrative review. But [7.Effects of multimorbidity and frailty on diabetes] is the result from the literature search.
Therefore, the reviewer recommend author to describe the method and the purpose more clearly.
And figure1 is fine as it is.
Author Response
Many thanks for your further comments. We have expanded on the methods section as possible and included subtitles for more clarification. Objectives are clearly described, we hope!